# Effect of Wet Aging on Color Stability, Tenderness, and Sensory Attributes of *Longissimus lumborum* and *Gluteus medius* Muscles from Water Buffalo Bulls

**DOI:** 10.3390/ani11082248

**Published:** 2021-07-30

**Authors:** Muhammad Hayat Jaspal, Iftikhar Hussain Badar, Osama Bin Amjad, Muhammad Kashif Yar, Muawuz Ijaz, Adeel Manzoor, Jamal Nasir, Bilal Asghar, Sher Ali, Kashif Nauman, Abdur Rahman, Um Ul Wara

**Affiliations:** 1Department of Meat Science and Technology, Faculty of Animal Production and Technology, University of Veterinary and Animal Sciences, Lahore 54000, Pakistan; osama.amjad@uvas.edu.pk (O.B.A.); adeel.rehman@uvas.edu.pk (A.M.); jamalnasir@uvas.edu.pk (J.N.); bilal.asghar@uvas.edu.pk (B.A.); sher.ali@uvas.edu.pk (S.A.); drkashif@uvas.edu.pk (K.N.); 2College of Food Science, Northeast Agricultural University, Harbin 150030, China; 3Department of Animal Sciences, CVAS-Jhang 35200, University of Veterinary and Animal Sciences, Lahore 54000, Pakistan; Kashif.yar@uvas.edu.pk (M.K.Y.); muawuz.ijaz@uvas.edu.pk (M.I.); abdurrehman@uvas.edu.pk (A.R.); 4Institute of Biochemistry and Biotechnology, University of Veterinary and Animal Sciences, Lahore 54000, Pakistan; umulwara197@gmail.com

**Keywords:** aging, metmyoglobin, sensory attributes, cooking loss, shear force value, myofibrillar fragmentation index

## Abstract

**Simple Summary:**

The water buffalo is found in many tropical countries worldwide. In the current world scenario, where meeting the protein requirements of the population is one of the biggest future challenges, buffalo meat could be a good source of protein and other nutrients. Currently, very little information is available regarding buffalo meat quality attributes. Therefore, this study was designed to evaluate the effects of aging time and muscle type on meat quality attributes (pH, color, tenderness, water holding capacity, and sensory acceptance) of buffalo meat. The results showed that color, tenderness, and sensory attributes were improved with aging time; the suitable aging time required to enhance meat quality attributes in *Longissimus lumborum* and *Gluteus medius* muscles is 28 and 21 days, respectively.

**Abstract:**

The present study aimed to investigate the effect of wet aging on meat quality characteristics of *Longissimus lumborum* (LL) and *Gluteus medius* (GM) muscles of buffalo bulls. Meat samples from six aging periods, i.e., 0 day (d) = control, 7 d, 14 d, 21 d, 28 d, and 35 d, were evaluated for pH, color, metmyoglobin content (MetMb%), cooking loss, water holding capacity (WHC), myofibrillar fragmentation index (MFI), Warner–Bratzler shear force (WBSF), and sensory evaluation. The pH, instrumental color redness (a *), yellowness (b *), chroma (C *), and MetMb% values were increased, while the lightness (L *) and hue angle (h *) values showed non-significant (*p* > 0.05) differences in both LL and GM muscles in all aging periods. The cooking loss increased while WHC decreased till 35 days of aging. MFI values significantly (*p* < 0.05) increased, while WBSF values decreased; in addition, sensory characteristics were improved with the increase in the aging period. Overall, the color, tenderness, and sensory characteristics were improved in LL and GM muscles until 28 and 21 days of aging, respectively. Based on the evaluated meat characteristics, 28 days of aging is required to improve the meat quality characteristics of LL, whereas 21 days of aging is suitable for GM muscle.

## 1. Introduction

The water buffalo (*Bubalus bubalis*) is found in many tropical countries globally, particularly in the Southeast Asia region [1]. These buffalo are primarily kept for milk purposes; however, they have excellent potential for meat production and play a vital role in the agricultural economies of many developing countries [2]. The water buffalo is an excellent converter of low-quality forage into good-quality meat, is resistant to many bovine diseases, and has excellent body weight gain, making it easier to manage them on locally available roughages [3]. Buffalo meat is almost similar to cattle meat; moreover, its meat has many superior characteristics, including higher protein, low fat, and cholesterol content [4]. In the current world scenario, where 1.4 billion people are protein deficient [5], buffalo meat could be a good source of protein and other nutrients, and it can serve a crucial part towards achieving the Sustainable Developmental Goal of the United Nations for food security for all.

Meat quality is described by many attributes, including its color, tenderness, juiciness, flavor, and palatability. Among all these attributes, tenderness is considered the primary quality characteristic regarding eating satisfaction [6,7]. This is evidenced by consumers’ willingness to pay the extra price for guaranteed tender beef [8]. Meat juiciness positively correlates with tenderness and depends on moisture retention during meat processing [9]. Inconsistency in meat tenderness has always been a significant concern for the beef industry. One reason for this inconsistency is the individual variations among different beef muscles [10,11]. Meat color is another critical quality attribute, as consumers associate bright cherry-red color with the freshness and wholesomeness of meat [12,13].

Researchers have developed various techniques to improve meat quality [14,15]. Post-slaughter interventions such as suspension methods, blade tenderization, exogenous enzymes, and conventional aging procedures are used to improve meat tenderness [16]. Among all these methods, postmortem aging is widely used by the industry to improve beef quality, particularly tenderness and palatability [8]. During postmortem aging, actin-myosin cross-linkages are broken down by different proteolytic enzymes that result in tender meat [17]. The aging process also improves meat color. The reduction in oxygen consumption rates with an enhanced storage period leads to the bright red color of meat [18]. Similarly, early activation of calpain-2 during the postmortem aging period coincides with improved meat’s juiciness [9]. Previous studies have performed the meat aging maximum until 28 days, and it is known that aging improves the tenderness during the first 14 days postmortem of cattle and lamb [19,20]. However, this study used the extended aging time (up to 35 days) because the proteolytic enzymes’ action is slower in buffalo than in cattle [7].

Among aging methods, wet aging has an advantage over dry-aging due to less weight loss, juicier meat, and little or no shrinkage of meat at storage temperature. This is also an inexpensive and less time-consuming method [18,21,22]. Being high value meat muscles among consumers, *Longissimus lumborum* (LL) and *Gluteus medius* (GM) were selected for the current study; as these muscles differ in their biochemical properties and tenderness, they respond differently to postmortem aging [8,23,24]. Therefore, the present study was designed to investigate the effect of wet aging on the meat quality characteristics of *Longissimus lumborum* (LL) and *Gluteus medius* (GM) muscles of buffalo bulls.

## 2. Materials and Methods

### 2.1. Ethics Statement

All studies related to animals were pre-approved (vide letter No. DR/567) by the Institutional Ethical Review Committee, Office of Research Innovation and Commercialization (ORIC), the University of Veterinary and Animal Sciences (UVAS), Lahore, Pakistan.

### 2.2. Animals Slaughtering

Buffalo bulls (*n* = 18) of the Nili-Ravi breed were procured from Research & Development Farm, Big Feed (Pvt.) Ltd., Lahore, Pakistan. Animals were 24 months of age (average body weight 290 ± 10 kg), reared under similar management and feeding conditions. All the animals were transported to the lairage facility of the University of Veterinary and Animal Sciences, Lahore, Pakistan. The animals were off-feed for 12 h before the slaughtering to ensure hygienic processing with free access to water. The animals were randomly slaughtered at University commercial slaughterhouse facility following the Halal slaughtering guidelines mentioned in Pakistan Halal Standards PS3733. The average hot carcass weights were 140 kg (SD = 10), and carcasses were chilled at 0–2 °C for 24 h.

### 2.3. Muscle Sampling and Aging Processing

Following chilling, carcasses were deboned at 10 ± 1 °C, and both-sided *Longissimus lumborum* (LL) and *Gluteus medius* (GM) muscles were removed. Every LL and GM was cut into steaks (*n* = 9), each steak with 2.5 cm thickness, and tagged for identification [7]. All LL (*n* = 18) steaks and GM steaks (*n* = 18) of each carcass were mixed separately for randomization. Three steaks were selected for each aging period (0 d = control, 7 d, 14 d, 21 d, 28 d, and 35 d) from both muscles of the carcass. One of the steaks was utilized to estimate pH, color, aging loss, cooking loss, tenderness, and water holding capacity (WHC), one was used for the evaluation of metmyoglobin and myofibrillar fragmentation index (MFI), and the remaining one steak was used for sensory evaluation by a sensory panel. Except for the control, all other steaks were vacuum packed in polyethylene bags (150 × 200, PA/PE 90) by a vacuum packing machine (Multivac^®^ Baseline, C300 twin, Wolfertschwenden, Germany) and stored at 0–2 °C for processing at 7 d, 14 d, 21 d, 28 d, and 35 d storage periods.

### 2.4. Study Parameters

#### 2.4.1. Instrumental Color

Instrumental color values of meat samples were measured using Minolta Chroma meter (Konica Minolta^®^ CR-410, Tokyo, Japan), provided with C illuminant, 2o standard observer, and 50 mm aperture, calibrated with a white tile each time before taking measurements [25]. Meat sample was removed from vacuum packs in the meat processing hall maintained at 10 ± 1 °C and bloomed for one hour in horizontal display chiller (ALVO, Model MD-12, size: 72″ × 42″ × 48″ by Technosight, Lahore, Pakistan) working at 0–4 °C [26]. Following the CIELAB color system, color values were measured, i.e., lightness (L *), redness (a *), yellowness (b *), hue angle (h), and chroma (C *). Three values from each steak were taken by placing the chroma meter at three different locations, and the mean value was calculated.

#### 2.4.2. pH Measurement

The pH of meat samples was measured by pH meter (WTW, pH 3210 SET 2, Wissenschaftlich-Technische Werkstätten GmbH, Weilheim, Germany), calibrated at pH 4 and 7. The pH meter probe was inserted at three different sites in the steak, and the mean value was calculated.

#### 2.4.3. Metmyoglobin Content

Metmyoglobin content (MetMb%) was determined as described by [27]. A meat sample of 5 g was homogenized with 25 mL of 40 mM phosphate buffer (pH 6.8) for 10 s and then refrigerated at 4 °C for one hour. Afterward, the sample was centrifuged at 4500× *g* for 30 min, and the supernatant was filtered by a Whatman filter paper No. 1, using a spectrophotometer. The sample’s absorption values were observed at wavelengths of 525, 545, 565, and 572 nm. The MetMb% of the meat sample was calculated with the given formula.
MetMb% = [−2.51 (A572/A525) + 0.777(A565/A525) + 0.8(A545/A525) +1.098] × 100(1)

In the above formula, A represents the absorbance value with respect to wavelengths.

#### 2.4.4. Water Holding Capacity (WHC)

Water holding capacity was measured with a compression machine (YYW-2, Nanjing Soil Instrument, Nanjing, China). A raw meat sample of 5 g was weighed and folded in filter papers. Then, it was compressed with a force of 373 N for 5 min. The weight of meat samples before and after the compression was recorded by the digital compact weighing balance (SF-400, 7000 g ± 1 g). The expressible water (%) was calculated using the following formula [28].
Expressible water (%) = w1 − w2/w1 × 100(2)
where w1 = initial weight of meat sample, and w2 = final weight of meat sample.

#### 2.4.5. Cooking Loss

For measuring cooking loss, samples were packed in polyethylene bags (150 × 200, PA/PE 90) and cooked in a waterbath (WNB45, Memmert GmbH + Co. KG, Schwabach, Germany) maintained at 82 °C until the core temperature of meat samples reached up to 72 °C [29] as recorded by digital food thermometer (TP101; temperature range of −50 °C to 300 °C). The weight of the initial raw meat sample and the cooked sample was recorded by digital compact weighing balance (SF-400, 7000 g ± 1 g, Zhejiang Tiansheng Electronic Co., Ltd., Zhejiang, China); the cooking loss was calculated using the following formula [30].
Cooking Loss (%) = w1 − w2/w1 × 100(3)
where w1 = weight of steak before cooking, and w2 = weight of steak after cooking.

#### 2.4.6. Warner–Bratzler Shear Force

The tenderness of meat samples was determined in terms of Warner–Bratzler shear force (WBSF) values. First, the cooked meat samples were chilled overnight at 0–4 °C [31]. Then, 6 cm long × 1 cm high × 1 cm wide strips of muscles were cut parallel to the direction of muscle fibers and sheared under a V-shaped stainless-steel blade, while the meat strips were placed with fibers oriented perpendicular to the Warner–Bratzler shear force device coupled with a texture analyzer (TA.XT plus^®^ texture analyzer, Stable Micro Systems Ltd., Godalming, UK). The WBSF value was recorded in newton per centimeter square (N/cm^2^). Three readings for each sample were taken. Furthermore, the mean value was calculated.

#### 2.4.7. Myofibrillar Fragmentation Index (MFI)

To estimate myofibrillar fragmentation index, a 50 mL solution was prepared containing cold sucrose (0.24 M) and potassium chloride (0.02 M). A minced meat sample of 10 g was added to the prepared solution and blended for 40 s at high speed in a tissue homogenizer. This homogenate was filtered using a cheesecloth, and after this, we also recorded the weight of the cheesecloth separately. The MFI was calculated as the residue’s weight in grams multiplied by 100 [32].
MFI = residue (grams) × 100(4)

#### 2.4.8. Sensory Analysis

The meat samples from different storage durations were taken and cooked on a hot plate to attain the core temperature of 80 °C. Each meat sample was divided into equal cubes and served to trained panelists comprised of 8 members, including postgraduate students, a lecturer, and an assistant professor from the University. The sensory analysis was performed in the sensory analysis lab at the Department of Meat Sciences UVAS Lahore, followed by the mouth rinse method to avoid the carry-over taste effect. To avoid biases, the tagging of samples was non-familiar to the sensory panel. Panelists judged the meat samples for overall tenderness, juiciness, flavor intensity, and overall acceptability on an 8-point hedonic scale, where 8 = extremely beef-like odor, extreme flavor intensity, extremely tender, extremely juicy, and high overall acceptance, while 1 = extremely non-beef-like odor, extremely weak flavor intensity, extremely tough, extremely dry, and extremely low overall acceptance [33].

### 2.5. Statistical Analysis

Data regarding the effect of aging period on meat color, tenderness, and sensory analysis were analyzed under a complete randomized design by using two-way ANOVA through SAS software (version 9.1). The significant difference was considered at *p* < 0.05. The means were compared using DMR Test [34].

The following mathematical model was used:Y = μ + F1i + F2j + (F1 F2)ij + εij(5)
where Y = response variable, μ = overall population mean, F1i = fixed effect of muscle type, F2j = fixed effect of aging period, (F1 F2)ij = interaction effect of muscle type and aging period, and εij = random error.

## 3. Results

### 3.1. Instrumental Color

The results of the instrumental color analysis after aging for 0, 7, 14, 21, 28, and 35 days are presented in Table 1.

Both muscle type and aging time significantly (*p* ≤ 0.05) influenced the CIE a * values; GM muscle had higher a * values than LL muscle. During the aging time, a significant (*p* < 0.05) increase in the CIE a * values was observed from 0 to 7 days, and the highest values were witnessed on days 7 and 14. Additionally, significant (*p* < 0.05) differences were found in muscle type and aging times’ interactions. Similarly, muscle type and aging time significantly (*p* < 0.05) affected the chroma values. Furthermore, muscle type and aging time interactions also showed significant (*p* < 0.05) differences in chroma values. The CIE b * values were significantly (*p* < 0.05) affected by the aging time and increased from day 0 to 35 days of the storage period. Non-significant differences were found in the values of CIE L * among the two muscles, the aging time, and their interactions. Likewise, no significant (*p* > 0.05) difference was observed in the hue values of the aging time and its interactions with muscle type; however, significant (*p* < 0.05) differences were found in the main effects of muscle type, and GM muscle had higher hue values than LL muscle.

### 3.2. pH

The results of this analysis are summarized in Table 1. There was no significant (*p* > 0.05) difference in pH of LL and GM muscles; however, the pH was significantly increased with the aging time. Muscle type and aging time interactions showed a non-significant effect on pH values.

### 3.3. Water Holding Capacity (WHC)

There were no significant (*p* > 0.05) differences in WHC of the two muscle types; however, aging time significantly (*p* < 0.05) affected the WHC% of both muscles. The WHC% decreased significantly with the increase in aging time (Figure 1). 

### 3.4. Cooking Loss

There was no significant (*p* > 0.05) interaction between the muscle type and aging time in cooking loss. Additionally, cooking loss values were independent of muscle type. However, significant (*p* < 0.05) differences were found in cooking loss values of aging times that increased significantly (*p* < 0.05) throughout the aging time (Table 2).

### 3.5. Warner–Bratzler Shear Force

As shown in Table 2, significant (*p* < 0.05) differences in shear force values were found in the main effect and interaction of muscle type and aging time. GM had lower shear force values than LL. The shear force values significantly (*p* < 0.05) decreased with an increase in aging time (0, 7, 14, 21, 28, and 35 days). The lowest shear force values were observed on day 35 in both GM and LL muscles.

### 3.6. Myofibrillar Fragmentation Index (MFI)

There was a significant (*p* < 0.05) difference in the main effect of the aging time, although a non-significant (*p* > 0.05) difference in MFI values of different muscles. The interactions between muscle type and aging time showed significant influence on MFI; both muscles presented an increasing trend with the aging time (Table 2).

### 3.7. Metmyoglobin (%)

As shown in Figure 2, non-significant (*p* > 0.05) differences were observed in the main effects of muscle type. At the same time, the metmyoglobin (%) values of these two muscles increased significantly (*p* < 0.05) with increased aging time. Similarly, the interactions between muscle type and aging period had significant (*p* < 0.05) effects on metmyoglobin (%), which increased with increasing aging duration in both muscle types.

### 3.8. Sensory Evaluation

The effect of muscle type, aging time, and their interactions on sensory evaluation is shown in Table 3. The LL muscle showed significantly (*p* ˂ 0.05) better overall acceptability than the GM muscle. However, the panelists observed non-significant differences in odor, flavor, tenderness, and juiciness between the two muscles. The meat texture and overall acceptability were significantly improved throughout the aging time. The meat texture and overall acceptability also showed significant interactions between muscle type and aging time. The LL muscle showed the highest texture characteristics and overall acceptability at 35 days of the storage period. 

## 4. Discussion

The present study aimed to evaluate the muscle-specific effects of aging time on the color, tenderness, and sensory attributes of buffalo meat. Generally, the L * values should be increased with increasing aging time due to protein denaturation with endogenous proteases, leading to the weakening of protein structure, resulting in more light scattering [7]. Similarly, the L * values of the LL and GM muscles increased linearly with an increase in aging time in the present study; however, this increase in L * values was statistically non-significant. Likewise, a * and C * values increased with aging time in both LL and GM muscles. The a * values of both muscles significantly increased up to 28 days of aging, while the C * values of LL and GM muscles increased up to 28 and 35 days, respectively. These results are inconsistent with the findings as reported by Stanišić et al. [35]. The authors found a significant increase in a * and C * values of LL and GM muscles during the aging periods.

Similarly, Stanišić et al. and Vitale et al. [35,36] found that the a * value of LL and GM muscles improved in the aged meat compared with non-aged meat. The improvement in meat color characteristics is attributed to the enhanced blooming ability of aged meat. Tang et al. and Jacob et al. [37,38] concluded that the difference between the blooming ability of the fresh un-aged meat cut surface and aged meat is due to a decrease in oxygen-consuming respiratory enzymes within the mitochondria of aged meat. This reduction allows more oxygen to bind with myoglobin, forming a thicker layer of oxymyoglobin [39]. Another possible reason for the increased a * value from 7 to 14 days of aging might be the exposure of the meat surface to oxygen, which penetrates the meat leading to the conversion of deoxymyoglobin to oxymyoglobin.

The pH values were gradually increased from 5.64 to 5.85 throughout the aging durations. In a similar study, Daszkiewicz et al. [40] also reported an increase in the pH of meat samples up to 14 days of aging. This increase in meat pH during aging could be attributed to the conformational changes linked with the proteolytic degradation of the muscle fibers [41] and hydrolysis of meat proteins into amino acids [42,43]. Likewise, Luz et al. [29] reported an increase in pH values up to 14 days of aging; however, afterward, the pH values decreased until 21 days of aging. This decrease in pH values between 14 and 21 days of aging might be due to a decrease in enzymatic activity with an extended aging time.

The WHC decreased with an increase in the aging time, while the cooking loss initially increased up to 7 days of aging and then decreased gradually throughout aging times. In their studies, Purslow et al. and Cho et al. [44,45] reported decreased WHC and cooking losses during different aging durations. The decrease in WHC with increasing aging duration is mainly caused by proteolysis and small protein fragments, leading to more fluid losses from muscle structure. When both muscles’ interactive effect was compared, LL had more drip losses than GM; this increase in drip losses could be due to more enzymatic reactions within the muscle structure [46]. Similarly, in the present study, an increase in the cooking loss up to 7 days of aging could be due to higher proteolysis during the earlier aging time. 

The tenderness of the meat is improved due to the proteolysis of myofibrillar proteins during aging [47,48,49]. Between two muscles, LL showed higher WBSF values than GM muscles; however, both muscles showed a significant decrease in the WBSF values and an increase in myofibrillar fragmentation index (MFI) values during an extended aging period. Similarly, various studies (Vaskoska et al., Shanks et al., Monsón et al., Koohmaraie et al., and Koohmaraie et al. [50,51,52,53,54]) reported a decrease in WBSF values and an increase in the MFI with the extended aging durations. The decrease in WBSF and increase in MFI values in GM muscle could be due to having low connective tissue content and more proteolytic breakdown of cross-linkages between muscle proteins in comparison to LL muscle [55]. In their study, Irurueta et al. [56] also noted a significant decrease in buffalo meat shear force values under vacuum storage of 25 days. This increase in the values of MFI over periods of aging is due to enhanced proteolysis by calpains resulting in myofibril fragmentation and leading to increased tenderness. This gives a fair indication that MFI can be used as an indicator of tenderness which could be used for selecting tender meat for processing and retail sale.

Similar to the present study, Mancini et al. [57] reported an increase in the metmyoglobin concentration with the increase in the aging periods. However, metmyoglobin is the indicator of the shelf life of meat color, and more metmyoglobin formation (brownish color meat) leads to a lower acceptance of the meat by the consumer. In the current study, although the metmyoglobin was increased with an increase in aging time, the improvement in the color characteristics with an increase in the aging period could be attributed to increased color intensities during the blooming process. To reduce metmyoglobin production, one should maintain the storage temperature (0–4 °C) and proper meat handling during transportation [58].

The sensory evaluation showed a better overall acceptability of LL in comparison GM muscle. This could be due to the comparatively better texture and juiciness of the LL than the GM muscle. The texture and overall acceptability were improved throughout the aging periods. Similarly, Monsón et al. [59] evaluated the effect of aging on the sensory characteristics of beef and observed better tenderness and overall acceptability with an aging period. The improvement in the sensory characteristics with aging time could be attributed to increased proteolysis and MFI with aging durations. Between the two muscles, LL showed the highest sensory texture and overall acceptability score throughout aging durations. This could be due to the highest MFI of LL throughout different aging durations, as reported in the current study.

## 5. Conclusions

Wet aging improved both muscles’ color attributes, tenderness, and sensory characteristics till 35 days of aging. Both LL and GM muscles responded differently to different aging periods. Based on the evaluated meat characteristics, overall, 28 days of aging is required to improve the meat quality of LL, while 21 days of aging is enough for improving the meat quality of GM muscle. However, further studies are needed to explore the effect of wet aging for extended periods on the microbial count, protein oxidation, and volatile compounds, along with the color stability and sensory attributes.

## Figures and Tables

**Figure 1 animals-11-02248-f001:**
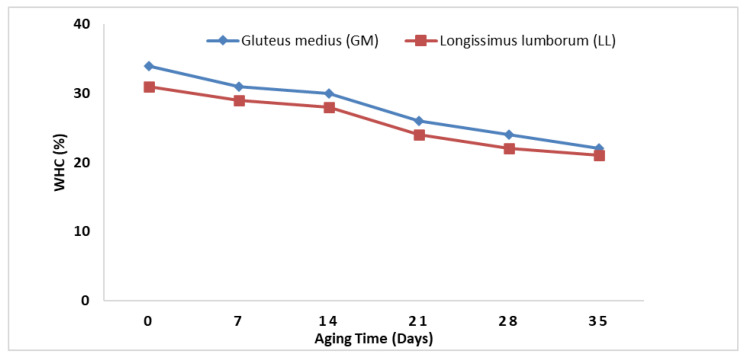
Effect of aging time on water holding capacity of two different muscles of buffalo meat.

**Figure 2 animals-11-02248-f002:**
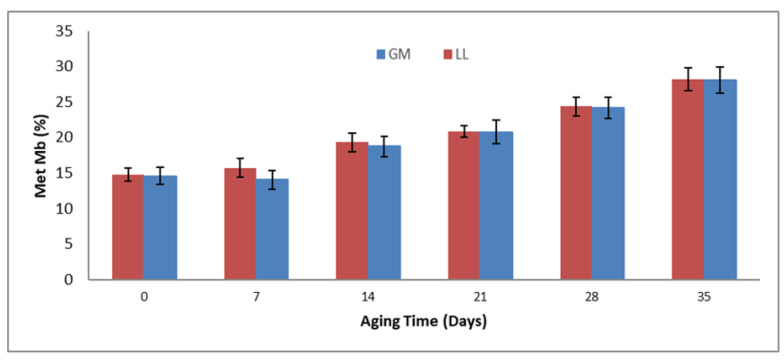
Effects of aging time and muscle type on Metmyoglobin% of buffalo meat.

**Table 1 animals-11-02248-t001:** Effects of aging time and muscle type on instrumental color (CIE L *, a *, b *, C *, h) and pH values of buffalo meat.

Aging (Days)	Muscle Type	Color					pH
		L *	a *	b *	C *	h	
0	*Longissimus lumborum* (LL)	43.15 ± 0.62	18.83 ± 0.40 ^d^	7.18 ± 0.27	19.96 ± 0.35 ^c^	20.73 ± 0.70	5.74 ± 0.07
7		44.65 ± 0.75	20.22 ± 0.45 ^abc^	8.21 ± 0.59	21.53 ± 0.55 ^abc^	20.86 ± 0.78	5.74 ± 0.04
14		45.11 ± 0.88	19.80 ± 0.44 ^abcd^	7.93 ± 0.30	22.38 ± 1.23 ^a^	20.93 ± 1.10	5.78 ± 0.05
21		44.13 ± 0.89	19.74 ± 0.36 ^bcd^	8.04 ± 0.52	20.73 ± 0.38 ^bc^	20.07 ± 0.82	5.81 ± 0.03
28		45.24 ± 0.79	21.07 ± 0.40 ^ab^	8.95 ± 0.46	21.19 ± 0.53 ^abc^	21.44 ± 0.84	5.84 ± 0.04
35		44.61 ± 0.37	19.53 ± 0.54 ^cd^	10.53 ± 1.83	20.91 ± 0.59 ^abc^	20.50 ± 0.71	5.84 ± 0.03
0	*Gluteus medius* (GM)	42.77 ± 0.55	19.59 ± 0.42 ^cd^	8.03 ± 0.55	20.83 ± 0.40 ^abc^	20.91 ± 0.91	5.65 ± 0.08
7		43.62 ± 0.86	21.17 ± 0.37 ^a^	9.22 ± 0.36	22.24 ± 0.52 ^ab^	22.38 ± 0.79	5.68 ± 0.04
14		43.78 ± 0.53	20.61 ± 0.45 ^abc^	9.27 ± 0.48	22.19 ± 0.60 ^ab^	22.86 ± 0.96	5.71 ± 0.06
21		44.26 ± 0.68	20.13 ± 0.34 ^abcd^	8.35 ± 0.50	21.36 ± 0.39 ^abc^	21.87 ± 1.38	5.75 ± 0.05
28		43.12 ± 1.80	20.34 ± 0.45 ^abc^	8.79 ± 0.43	21.80 ± 0.57 ^abc^	21.79 ± 0.66	5.78 ± 0.03
35		44.81 ± 0.80	20.49 ± 0.45 ^abc^	9.12 ± 0.57	22.48 ± 0.60 ^a^	23.29 ± 1.04	5.82 ± 0.03
*p*-value							
Muscle type Aging (days) Interactions		0.133 0.438 0.739	0.034 0.005 0.006	0.422 0.048 0.110	0.030 0.024 0.048	0.007 0.762 0.701	0.5451 0.0236 0.2302

^a,b,c,d^ Superscripts on different means within column differ significantly, *p* ≤ 0.05; L * = Lightness, a * = redness, b * = yellowness, C * = chroma, h = hue angle.

**Table 2 animals-11-02248-t002:** Effect of aging time and muscle type on cooking loss, WBSF, and MFI values of buffalo meat.

Aging (Days)	Muscle Type	Parameters		
		Cooking Loss	WBSF	MFI
0	*Longissimus lumborum* (LL)	35.67 ± 1.04	49.85 ± 3.14 ^a^	567.78 ± 30.55 ^c^
7		38.19 ± 1.05	43.38 ± 2.45 ^ab^	588.39 ± 37.12 ^c^
14		37.24 ± 0.86	37.47 ± 2.28 ^bc^	634.83 ± 33.39 ^bc^
21		37.56 ± 0.69	34.37 ± 1.69 ^cd^	649.44 ± 41.47 ^abc^
28		36.12 ± 1.16	29.12 ± 1.44 ^def^	653.22 ± 51.97 ^abc^
35		36.26 ± 1.30	24.75 ± 1.81 ^f^	768.44 ± 34.32 ^a^
0	*Gluteus medius* (GM)	36.22 ± 0.71	47.81 ± 3.28 ^a^	544.56 ± 47.41 ^c^
7		38.40 ± 0.50	38.58 ± 2.73 ^bc^	568.83 ± 36.15 ^c^
14		38.04 ± 0.43	33.48 ± 2.63 ^cde^	588.06 ± 29.05 ^c^
21		37.85 ± 0.79	29.17 ± 1.90 ^def^	593.11 ± 52.05 ^c^
28		36.15 ± 1.19	27.03 ± 1.77 ^ef^	650.78 ± 43.91 ^abc^
35		36.90 ± 1.04	25.96 ± 1.77 ^f^	736.83 ± 40.31 ^ab^
*p*-value				
Muscle type Aging (days)		0.2969 0.0104	0.0361 <0.0001	0.2011 <0.0001
Interactions		0.0609	<0.0001	0.0016

^a,b,c,d,e,f^ Superscripts on different means within column differ significantly, *p* ≤ 0.05; WBSF = Warner–Bratzler Shear force, MFI = Myofibrillar Fragmentation Index.

**Table 3 animals-11-02248-t003:** Effect of aging time and muscle type on sensory characteristics of Buffalo meat.

Aging (Days)	Muscle Type	Odor	Flavor Intensity	Overall Tenderness	Juiciness	Overall Acceptability
0	*Longissimus lumborum* (LL)	6.12 ± 0.45	5.78 ± 0.60	6.25 ± 0.73 ^f^	5.48 ± 0.61	5.42 ± 0.62 ^c^
7		6.08 ± 0.50	5.75 ± 0.53	6.66 ± 0.57 ^e^	5.54 ± 0.59	5.44 ± 0.59 ^c^
14		6.05 ± 0.19	6.69 ± 0.56	6.97 ± 0.65 ^d^	5.59 ± 0.81	5.50 ± 0.81 ^c^
21		6.00 ± 0.26	5.72 ± 0.72	7.32 ± 0.69 ^c^	5.65 ± 0.73	5.67 ± 0.56 ^b^
28		5.97 ± 0.35	5.76 ± 0.51	7.47 ± 0.67 ^b^	5.64 ± 0.83	5.69 ± 0.28 ^b^
35		5.99 ± 0.31	6.82 ± 0.42	7.86 ± 0.40 ^a^	5.66 ± 0.52	5.79 ± 0.57 ^a^
0	*Gluteus medius* (GM)	6.06 ± 0.29	5.69 ± 0.72	6.31 ± 0.89 ^f^	5.31 ± 0.90	5.24 ± 0.78 ^e^
7		6.03 ± 0.49	5.62 ± 0.82	6.61 ± 0.69 ^e^	5.47 ± 0.58	5.32 ± 0.71 ^d^
14		5.99 ± 0.66	5.69 ± 0.54	7.02 ± 0.80 ^d^	5.48 ± 0.80	5.45 ± 0.39 ^c^
21		5.95 ± 0.11	5.72 ± 0.32	7.20 ± 0.16 ^c^	5.56 ± 0.50	5.51 ± 0.21 ^c^
28		5.90 ± 0.39	6.75 ± 0.50	7.54 ± 0.42 ^b^	5.53 ± 0.63	5.56 ± 0.43 ^c^
35		5.89 ± 0.19	6.70 ± 0.27	7.63 ± 0.39 ^ab^	5.54 ± 0.63	5.51 ± 0.45 ^c^
*p*-value						
Muscle type Aging (days)		0.4692 0.0321	0.0221 0.8030	0.8322 0.6823	0.6726 0.9639	0.0182 0.0251
Interactions		0.8534	0.9353	0.0212	0.9990	0.0158

^a,b,c,d,e,f^ Superscripts on different means within column differ significantly, *p* ≤ 0.05; Odor score: 8 = extremely beef like, 1 = extremely non-beef like. Flavor score: 8 = extremely strong, 1 = extremely weak. Texture score: 8 = extremely tender, 1 = extremely tough. Juiciness score: 8 = extremely juicy, 1 = extremely dry. Overall acceptability score: 8 = extremely acceptable, 1 = extremely unacceptable.

## Data Availability

The data are not publicly available.

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
