# Peer review of "Effect of Wet Aging on Color Stability, Tenderness, and Sensory Attributes of Longissimus lumborum and Gluteus medius Muscles from Water Buffalo Bulls"

_animals, 2021, doi:10.3390/ani11082248_

Round 1
Reviewer 1 Report
see attached file

Author Response
Dear Reviewer,
Thank you for your time in reviewing our manuscript. We have carefully studied the comments, modified the manuscript according to your comments and suggestions. Your comments helped us to improve this manuscript.
Thanks and best regards,
Iftikhar

Reviewer 2 Report
The manuscript submitted by Jaspal et al. deals with the evaluation of the Effects of wet aging on color stability, tenderness, and sensory attributes of longissimus lumborum and gluteus medius muscles from water buffalo bulls. The work is quite original, the language is clear and the applied methods adequately described. However there are two major points that need to be addressed before proceeding further.
First of all, in lines 97-98 authors stated that: “…reared under similar management and feeding conditions.” what aspects differ? this statement if not adequately explained could represent a problem for the scientific solidity of the work. The diet, as well as the breeding strategy have a huge influence on the quality of animal products, so for the structure of the work I believe the presence of differences in the management of the animals could represent a significant limit.
In addition to this it must be said that meat has been subjected to basic evaluations mostly of a physical nature and the study would be much better if some nutritional parameters were also evaluated (total protein, fat, fatty acid profile, oxidative stability, ecc.).
Minor concerns:
L187: How many panelists?
Figure 1b should be modified. The abscissa axis must be lengthened for a clearer display of the data.
In Fugures 1b and 2a the units of measurement must be indicated on the ordinate axis.
Author Response

(The authors gave the same response as above.)

Round 2
Reviewer 1 Report
The authors responded accurately to the suggestions and changes indicated. These changes have improved the manuscript, especially the results section with the improved tables and figures.
Minor comments:
In the discussion section, authors should add the first author´s last name followed by et at. , when it is requested. See lines 289, 291, 293, 301, 304, and so far. An example: in line 289 it should be written: “…..findings as reported by Stanišić et al. [37].”
Line 27. The term tenderness should be changed to “Warner Bratzler shear force” or instrumental tenderness. Shear force is an instrumental (objective) measurement of tenderness. The term tenderness (muscle fiber tenderness or overall tenderness) usually refers to sensory or palatability traits.
Line 29: “insignificant” is similar to “no significant” in terms of ANOVA results. Also, see line 214.
Line 32-34. Conclusion in the abstract should be revised with the modified conclusion (see lines 352-354)
In Table 2. “shear force” should be changed to “ WBSF”
In Table 3 change intesity by “intensity”
Author Response
Thank you for your comments, appreciation, and kind words. Your comments helped us to improve this manuscript for possible publication in Animals.
Best regards,
Iftikhar

Reviewer 2 Report
The criticisms raised during the first round of revision have been adequately addressed.
Author Response

(The authors gave the same response as above.)
